

# The effects of subjective family status and subjective school status on depression and suicidal ideation among adolescents: the role of anxiety and psychological resilience

Zhan Shu*, Shurui Chen*, Hui Chen, Xianliang Chen, Huajia Tang, Jiawei Zhou, Yusheng Tian, Xiaoping Wang and Jiansong Zhou

National Clinical Research Center for Mental Disorders, and Department of Psychiatry, The Second Xiangya Hospital, Central South University, Changsha, China
* These authors contributed equally to this work.

Corresponding authors
Xiaoping Wang, xiaop6@csu.edu.cn
Jiansong Zhou,
zhoujs2003@csu.edu.cn

## ABSTRACT

**Background:** Depression and suicidal tendencies are notably prevalent among adolescents, yet few studies have revealed the impact of social status on them. This study aimed to explore the mediating and moderating effects of anxiety and psychological resilience on family status, school status, depression, and suicidal ideation.

**Methods:** A total of 1,190 secondary school students aged 13 to 17 years (mean age: 13.57 ± 2.02 years) were evaluated depression, anxiety, and suicidal ideation using the PHQ-8 and GAD-7 questionnaires. Subjective family and school status were also assessed. Data analysis was conducted using Mplus, SPSS's Process, and the RSA 3.0 plugin.

**Results:** Subjective family/school status, anxiety, psychological resilience, depression, and suicidal ideation were significantly correlated. Anxiety played a partial mediating role in the influence of subjective family status and subjective school on depression and suicidal ideation, and psychological resilience moderated the impact of anxiety on adolescent depression and suicidal ideation ($\beta_{depression}$ = −0.05, $p < 0.01$; $\beta_{suicidal\ ideation}$ = −0.06, $p < 0.05$).

**Conclusions:** Subjective family status and school status played important roles in depression and suicidal ideation in adolescents, and anxiety and psychological resilience played mediating and moderating roles.

## INTRODUCTION

Depression and suicidal tendencies are of particular concern due to their high prevalence and serious outcomes (*Trivedi et al., 2023*). In 2020, the incidence of depression among Chinese teenagers was 24.6%, with 7.4% exhibiting severe depression (*Fu & Zhang, 2020*). Moreover, suicide is now the second leading cause of death among adolescents aged 15 to 19 years (*Breslin, Balaban & Shubkin, 2020*; *Thompson & Swartout, 2018*). Suicidal ideation is a risk factor for suicide attempts and suicidal behavior, with a significant

predictive role in suicide. Adolescence, as a critical period for suicide prevention and intervention, has been extensively studied and is of great concern to researchers (*Zhu, Tian & Huebner, 2019*). Therefore, jointly investigating the developmental mechanisms of depression and suicidal ideation among adolescents is necessary.

In recent years, numerous studies have investigated risk factors for depression and suicidal ideation, highlighting the pivotal role of social factors. Social status is a crucial aspect of social factors and typically denotes an individual's position in the social hierarchy, such as wealth, education level, occupation, and social status (*Upenieks, Schieman & Meiorin, 2022*). Some studies have indicated that people with lower social status are more prone to depression and suicidal ideation (*Dickerson, Milojevich & Quas, 2022*; *Rivenbark et al., 2020*; *Vidal & Latkin, 2020*). Furthermore, the social causation hypothesis posits that individuals with low social status experience significantly higher risks of depression and suicide than those with high social status (*Dohrenwend & Dohrenwend, 1970*). This could be attributed to the fact that individuals with lower social status encounter more stress and obstacles, such as poverty, discrimination, and social exclusion. These experiences may increase anxiety and helplessness, which heightens the risk of depression and suicidal ideation (*Bøe et al., 2019*).

On the one hand, low subjective socioeconomic status increases the risk of depression by intensifying feelings of despair and loss of control (*Lund et al., 2011*; *Pössel, Wood & Roane, 2022*). On the other hand, research has suggested that having sufficient psychological resources can help individuals to cope with stress, alleviating the adverse effects of negative emotions and promoting personal growth (*Masten, 2014*). Adolescents who possess sufficient psychological resources are capable of achieving positive psychological development even when experiencing high levels of anxiety (*Masten, 2014*; *Wright, Masten & Narayan, 2013*). Therefore, this study posits that mental resources, such as psychological resilience, may mitigate the impact of anxiety on depression and regulate anxiety in adolescents.

## The effects of subjective family status and subjective school status on depression and suicidal ideation in adolescents

A large body of empirical research has demonstrated that subjective social status has a negative impact on adolescents' depression and suicidal ideation (*Dickerson, Milojevich & Quas, 2022*; *Upenieks, Schieman & Meiorin, 2022*; *Madigan & Daly, 2023*). Bettina and Elizabeth suggested that low subjective family socioeconomic status may have a negative impact on depression (*Chen & Paterson, 2006*). Goodman and Huang used subjective family socioeconomic status to predict mental health issues such as suicide (*Markham et al., 2007*). Studies have also used total scores of both family and school socioeconomic status. For example, several studies have found that the total score of socioeconomic status can negatively predict depression (*Adler et al., 2008*; *Demakakos et al., 2008*). However, the shared and unique effects of subjective family status and subjective school status on depression and suicidal ideation among adolescents according to subjective economic status remain unclear. Therefore, the second aim of this study was to explore four situations varying in the congruency of subjective family status and subjective school status
among adolescents: (1) the "safe" status (high family status and high school status), (2) the "high-risk" status (low family status and low school status), (3) the "high-family" status (high family status and low school status) and (4) the "low-family" status (low family status and high school status). Analysis of congruency, such as polynomial regression and response surface analysis, can provide a more in-depth and detailed characterization of the complex relationship between the two variables. Therefore, this study used polynomial regression and response surface analysis to thoroughly examine the joint effects of subjective family status and subjective school status on anxiety among adolescents. Furthermore, we proposed the following hypothesis (H1): Subjective family status and subjective school status affect anxiety among adolescents.

## The influences of subjective family status and subjective school status on anxiety and its mediating effect

Anxiety seriously affects the physical and mental health of individuals and has consistently attracted attention (*Beharry, 2022*; *Madigan & Daly, 2023*). The diathesis-stress model suggests that external stressors can interact with an individual's physiological vulnerability or predisposition, leading to the manifestation of psychological symptoms such as depression and suicidal behaviors (*Rosenthal, 1970*). Based on this model, research has found that adolescents, being individuals in a susceptible period of psychological development, can be negatively impacted by external stressors such as low social and economic status. These stressors have joint effects on susceptibility and can lead to the development of anxiety (*Jia & Zhu, 2021*). There's also research that shows subjective economic status has a negative impact on anxiety. For instance, *Zvolensky et al. (2016)* found that individuals with higher subjective socioeconomic status during childhood experienced lower levels of negative emotions such as anxiety and depression. *Kraus et al. (2012)* argued that individuals with lower socioeconomic status exhibit higher levels of anxiety than those with higher socioeconomic status. They proposed that anxiety, as a product of stress sensitivity, is one of the inherent characteristics of individuals in lower socioeconomic classes (*Kraus et al., 2012*). *Copeland et al. (2009)* conducted a longitudinal study and found that feelings of anxiety in adolescents can predict feelings of depression. In another longitudinal study involving 816 youths, the combination of early exposure to stressors and a previous history of anxiety disorder predicted an increase in depressive symptoms when exposed to low levels of current stress (*Copeland et al., 2009*). Therefore, we proposed that subjective family and school statuses influence depression and suicidal ideation through anxiety (*i.e.*, that anxiety plays a mediating role). Thus, our second hypothesis (H2) is as follows: subjective family status and school status influence adolescent depression and suicidal ideation through anxiety.

Additionally, the stress-buffering model suggests that depression and suicidal ideation are not inevitable consequences of anxiety and that the relationship between the two is moderated by other external factors (*Menéndez-Aller et al., 2020*; *Zhang & Zhou, 2018*; *Chen & Wang, 2013*). Therefore, the third aim of this study was to investigate moderating effects in this mediating model.

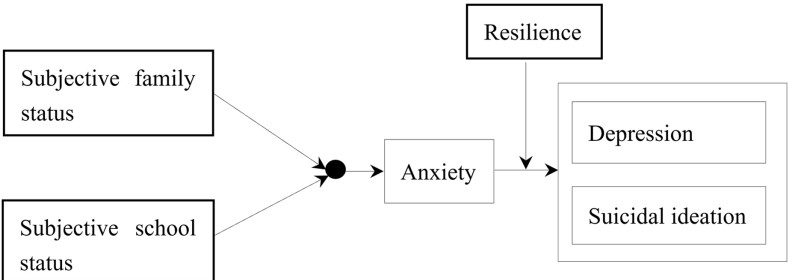

**Figure 1 Theoretical hypothesis model.** (1) Polynomial regression and response surface analysis can reveal the impact of adolescents' subjective family status and subjective school status on anxiety, (2) subjective family status and school status influence adolescent depression and suicidal ideation through anxiety, and (3) psychological resilience can regulate the effects of anxiety on depression and suicidal ideation in adolescents.

## Moderating effect of psychological resilience

The diathesis-stress model of resilience suggests that individuals with resilience are less likely to experience psychological disorders and diseases when faced with stress (*O'Connor & Kirtley, 2018*). Resilience can help buffer the impact of negative emotions on depression and suicidal ideation (*Hoorelbeke et al., 2019*; *Konradt et al., 2018*). Psychological resilience can alleviate the negative effects of adverse events and consequences, serving as a protective factor that improves an individual's mental health and well-being (*Navrady et al., 2018*). When facing adversity events, individuals can alleviate negative effects by increasing their psychological toughness, thus reducing the risk of psychological disorders (*Imran et al., 2022*). Research on adolescents has shown that individuals can reduce their depressive and suicidal thoughts by improving their psychological resilience (*Ai & Hu, 2016*). *Cong et al. (2019)* and *Hirschtritt et al. (2015)* confirmed that improving the psychological resilience of adolescents can reduce suicidal ideation. Furthermore, the results of intervention studies targeting depression and suicidal ideation have shown that psychological resilience buffer the negative effects of these conditions (*Konradt et al., 2018*; *Cong et al., 2019*). Psychological resilience can reduce the incidence of depression and suicidal ideation (*Hoorelbeke et al., 2019*; *Navrady et al., 2018*; *Hiyoshi et al., 2017*). Thus, we predicted that psychological resilience plays a moderating role in the latter part of this mediating model, specifically moderating the impact of anxiety on depression and suicidal ideation in adolescents.

In conclusion, building upon previous studies, this research aimed to explore the developmental mechanisms of depression and suicidal ideation in adolescents and proposed a moderated mediation model (as shown in Fig. 1). This study had three main assumptions: (1) polynomial regression and response surface analysis can reveal the impact of adolescents' subjective family status and subjective school status on anxiety, (2) subjective family status and school status influence adolescent depression and suicidal ideation through anxiety, and (3) psychological resilience can regulate the effects of anxiety on depression and suicidal ideation in adolescents.

## MATERIALS AND METHODS

### Participants and procedures

This was a cross-sectional study conducted from August 2021 to October 2021. Using a cluster sampling by class, 1,190 students were recruited from a secondary school in Changsha, China. The Chinese online questionnaire platform Wenjuanxing (https://www.wjx.cn/) was utilized to collect information and responses from all participants, with the link being sent to parents. After parents understood the study's purpose and agreed to participate, they clicked the "Agree" button online, and students completed the questionnaire accompanied by their parents. Students were eligible if they were aged between 13 and 17 years and fully understood the questionnaire's content. During the testing period, participants were informed that they had the right to withdraw from the test at any time due to discomfort. Students could raise questions about the questionnaire, which were answered online by staff. After the measurements, staff randomly selected a few students from each class to inquire about their understanding of the questions and whether they experienced any negative impact. The survey results showed that participants were able to accurately understand the questions and did not experience any negative psychological effects. The exclusion criteria were refusal to participate in the investigation or failure to complete the questionnaire carefully. The data for this study comes from a part of the Chinese Depression Cohort Study (CDCS), with participants being students from a middle school in Hunan, China. The total number of participants was 1,258, with 68 missing values (5.4%). These 68 participants were excluded due to missing variables and incomplete data. Among the participants, there were 595 in the first year of junior high, 550 in the second year, and 45 in the third year. This study was approved by the Ethics Committee of the Second Xiangya Hospital of Central South University (2018S007).

### Instruments

#### Subjective family status and subjective school status

To measure subjective family status and subjective school status, two questions were used; these questions asked participants whether they perceived their family or school to be positioned at the top or bottom in relation to other families or schools, respectively (*Goodman et al., 2007*). Respondents were asked to position themselves on a scale that ranged from 10 (top) to bottom (1) using labeled rungs on a ladder (*Evans & Kelley, 2004*).

#### Anxiety

The Chinese version of the General Anxiety Disorder-7 scale (GAD-7) (*Spitzer et al., 2006*) was utilized to assess anxiety symptoms experienced within the last 2 weeks. The GAD-7 consists of seven items that are rated on a four-point Likert scale, ranging from 0 (not at all) to 3 (nearly every day). Previous studies have demonstrated that the GAD-7 has good reliability and validity (*He et al., 2010*), and in this study, the Cronbach's alpha coefficient was 0.911, indicating high internal consistency.

### Depression and suicidal ideation

The eight-item Patient Health Questionnaire (PHQ-8) (*Dhingra et al., 2011*) is a self-assessment scale used to measure depressive symptoms over the previous 2 weeks in various populations, including adolescents (*Wu et al., 2020*). The scale consists of items that are rated on a four-point Likert scale ranging from 0 (not at all) to 3 (nearly every day), with higher total scores indicating more severe depressive symptoms. In this study, the PHQ-8 demonstrated good reliability and validity with a Cronbach's alpha of 0.892.

Based on previous research (*Kroenke et al., 2010*; *Lawrence et al., 2010*; *Walker et al., 2008*), the assessment of suicidal ideation in this study involved the question: "Have you thought about dying or hurting yourself in any way in the past 2 weeks?" Participants' responses ranged from 1 (no) to 4 (almost every day).

## Statistical analysis

Confirmatory factor analysis and model fitting were conducted using Mplus 7.4. The SPSS plug-in RSA 3.0 was utilized to compute the slope and curvature of feature lines, and Pearson correlation analysis was used to examine the correlations between variables, while SPSS 24.0 and the SPSS macro-PROCESS developed by Hayes were employed to test the mediation models. All significance tests of the regression coefficients were performed using the bootstrap method (*Bai et al., 2020*).

Polynomial regression was employed to investigate the effects of congruency of subjective family status and school status on adolescents' anxiety. These methods have been demonstrated to be reliable and informative for analyzing congruency effects in previous research (*Edwards & Parry, 1993*; *Shanock et al., 2010*) and have been widely utilized in empirical studies (*Laird & De Los Reyes, 2013*; *Nelemans et al., 2016*; *Pierce, Zhdanova & Lucas, 2017*; *Song, Yang & He, 2021*; *Weidmann et al., 2017*). Subjective family status and subjective school status were centered around the pooled grand mean to minimize multicollinearity before computing the second-order terms, following the procedure outlined by *Edwards & Parry (1993)*. Adolescents' anxiety was regressed on control variables, centered subjective family status (F), centered subjective school status (S), family squared ($F^2$), family status times school status ($F \times S$), and school status ($S^2$). The model was as follows (control variables omitted for simplicity): Anxiety = b0 + b1F + b2S + b3F$^2$ + b4S$^2$ + b5F $\times$ S + e. In which F represents subjective family economic status, S represents subjective school economic status, F $\times$ S is the interaction term between the two, along with their squared terms; $b_0$ represents the intercept, $b_1$ is the coefficient for F, $b_2$ is the coefficient for S, $b_3$ is the coefficient for F squared, $b_4$ is the coefficient for the interaction term, $b_5$ is the coefficient for S squared, and e is the error term. This article first standardizes the measurement indicators F and S in terms of scale, then regresses each item, and presents the results by plotting a three-dimensional graph. In the three-dimensional graph, we mainly determine the impact on the outcome variable by calculating the slope $a_1 = b_1 + b_2$ and curvature $a_2 = b_3 + b_4 + b_5$ of the "F = S" matching curve, as well as the slope $a_3 = b_1 - b_2$ and curvature $a_4 = b_3 - b_4 + b_5$ of the "F = −S" mismatching curve, and their significance (*Shanock et al., 2010*). We used regression coefficients to create three-dimensional response surfaces, with F and S on the

**Table 1  Sociodemographic characteristics.**

|  |  | Frequency | Percent (%) |
|---|---|---|---|
| Gender | Male | 546 | 45.9 |
|  | Female | 644 | 54.1 |
| Grade | 7 | 595 | 50 |
|  | 8 | 550 | 46.2 |
|  | 9 | 45 | 3.8 |

|  | Mean | SD |
|---|---|---|
| Age | 13.57 | 2.02 |
| Subjective family status | 6.72 | 2.08 |
| Subjective school status | 6.67 | 2.08 |
| GAD score | 2.25 | 3.84 |
| Resilience | 46.95 | 27.74 |
| PHQ score | 2.57 | 3.84 |
| Suicidal ideation | 0.21 | 0.57 |

**Note:**
Demographic characteristics and clinical variables ($N$ = 1,190).

perpendicular horizontal axes and anxiety on the vertical axis. To test for a moderated mediation effect, polynomial regression coefficients were used to create a block variable, which represented subjective family status and subjective school status (*Edwards & Cable, 2009*). The moderated mediation effect was then tested using the block variables as independent variables. A moderated mediating effect method was used to examine the influence of block variables, including subjective family status and subjective school status, on adolescent depression and suicide attempts.

The study further investigated the mediating effect of anxiety and the moderating effect of psychological resilience on the mediating effect. To accomplish these goals, the study constructed three models of the relationships among these variables. Model 1 estimated the prediction of the independent block variable on the dependent variables of adolescent depression and suicide attempts. Model 2 estimated the prediction of the block variable on the intermediate variable anxiety. Model 3 estimated the moderating effect of psychological resilience on the relationships among anxiety, depression, and suicide attempts in adolescents and tested the residual effect of the block variable on depression and suicide attempts in adolescents. All continuous variables were standardized.

# RESULTS

## Sociodemographic characteristics
The study enrolled 1,190 Chinese middle school students, comprising 546 boys and 644 girls, who were aged 13 to 17 years (mean age of 13.57 ± 2.02 years) (see Table 1).

## Correlation analysis
Table 2 lists the correlation results of each research variable. Only age and school status had no significant correlation, while the other variables (control variables, subjective

**Table 2 Results of correlation analysis between variables.**

| Variables | 1 | 2 | 3 | 4 | 5 | 6 | 7 | 8 |
|---|---|---|---|---|---|---|---|---|
| Sex | — | | | | | | | |
| Age | −0.017 | — | | | | | | |
| Subjective family status | 0.056 | 0.008 | — | | | | | |
| Subjective school status | −0.042 | 0.001 | 0.550** | — | | | | |
| Anxiety | 0.121*** | −0.002 | −0.131*** | −0.144*** | — | | | |
| Resilience | −0.009 | −0.037 | 0.185** | 0.256*** | −0.022 | — | | |
| Depression | 0.159*** | −0.01 | −0.166** | −0.177*** | 0.817*** | −0.066* | — | |
| Suicidal ideation | 0.110** | −0.008 | −0.176*** | −0.196*** | 0.563*** | −0.108*** | 0.635*** | — |

Notes:
* $p < 0.05$.
** $p < 0.01$.
*** $p < 0.001$.

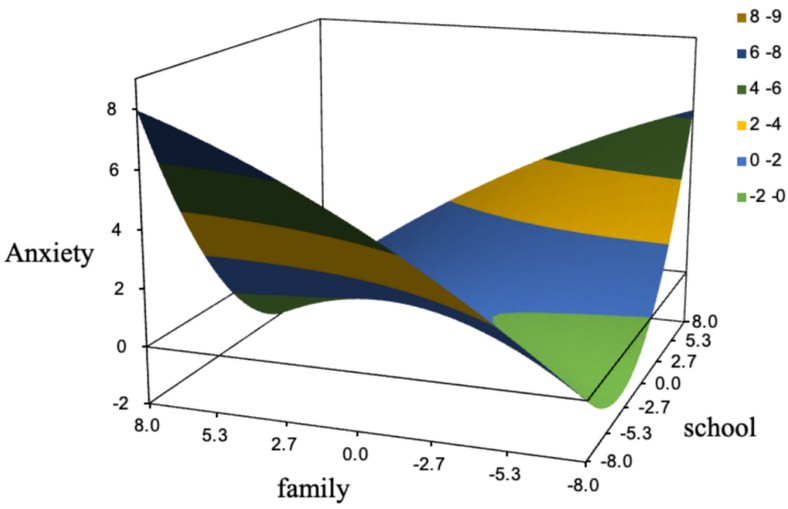

**Figure 2 The effect of subjective family status—subjective school status uniformity on adolescent anxiety.** To examine the effects of congruency of subjective family status and school status on adolescent anxiety, response surface analysis was conducted.

family/school status, anxiety, psychological resilience, depression, suicidal ideation) had significant correlations.

## The effect of subjective family status and subjective school status on adolescent anxiety

To examine the effects of congruency of subjective family status and school status on adolescent anxiety, response surface analysis was conducted. Prior to the analysis, the sample response ratio was calculated to determine its suitability for polynomial regression and response surface analysis. The findings indicated that after standardizing the scale, 60.75% (723) of the samples had a subjective family economic status consistent with the subjective school economic status, while 17.98% (214) had a subjective family economic status greater than the school economic status, and 21.26% (253) had a subjective family economic status lower than the school economic status. To meet the analysis requirements

**Table 3 Model of moderated mediating effect (Dependent variable Y1: depression).**

| | Model 1 (Dependent variable: Y1) | | | Model 2 (Dependent variable: M) | | | Model 3 (Dependent variable: Y1) | | |
|---|---|---|---|---|---|---|---|---|---|
| | β | t | 95% CI | β | t | 95% CI | β | t | 95% CI |
| Gender | 0.16*** | 5.61 | [0.10–0.21] | 0.12*** | 4.34 | [0.07–0.18] | 0.06*** | 3.38 | [0.02–0.09] |
| Age | −0.01 | −0.22 | [−0.06 to 0.05] | 0.01 | 0.05 | [−0.05 to 0.06] | −0.008 | 0.65 | [−0.0 to 0.02] |
| X | 0.20*** | 7.13 | [0.14–0.25] | 0.16*** | 5.70 | [0.10–0.22] | 0.06*** | 3.83 | [0.03–0.10] |
| M | | | | | | | 0.79*** | 46.56 | [0.76–0.83] |
| W | | | | | | | −0.04* | −2.32 | [−0.07 to −0.01] |
| MW | | | | | | | −0.05** | −2.46 | [−0.08 to −0.01] |
| $R^2$ | 0.63 | | | 0.04 | | | 0.68 | | |
| F | 27.68*** | | | 10.57*** | | | 357.71*** | | |

Note:
X is a block variable of subjective family economic status and subjective school status; M is the mediator variable anxiety; W is the moderator variable resilience; the dependent variables were adolescent depression (Y1), NSSI (Y2).
* $p < 0.05$.
** $p < 0.01$.
*** $p < 0.001$.

**Table 4 Model of moderated mediating effect (Dependent variable Y2: suicidal ideation).**

| | Model 1 (Dependent variable: Y2) | | | Model 2 (Dependent variable: M) | | | Model 3 (Dependent variable: Y2) | | |
|---|---|---|---|---|---|---|---|---|---|
| | β | t | 95% CI | β | t | 95% CI | β | t | 95% CI |
| Gender | 0.11*** | 3.84 | [0.0–0.16] | 0.12*** | 4.34 | [0.07–0.18] | 0.04 | 1.47 | [−0.01 to 0.08] |
| Age | −0.01 | −0.20 | [−0.06 to 0.05] | 0.001 | 0.05 | [−0.06 to 0.06] | −0.01 | −0.34 | [−0.05 to 0.04] |
| X | 0.19*** | 7.02 | [0.14–0.25] | 0.16*** | 5.70 | [0.11–0.22] | 0.09*** | 3.77 | [0.04–0.14] |
| M | | | | | | | 0.53*** | 21.89 | [0.49–0.58] |
| W | | | | | | | −0.08* | −3.38 | [−0.13 to −0.04] |
| MW | | | | | | | −0.06* | −2.27 | [−0.11 to −0.01] |
| $R^2$ | 0.05 | | | 0.04 | | | 0.34 | | |
| F | 21.49*** | | | 10.57*** | | | 87.15*** | | |

Note:
X is a block variable of subjective family economic status and subjective school status; M is the mediator variable anxiety; W is the moderator variable resilience; the dependent variables were adolescent depression (Y1), NSSI (Y2).
* $p < 0.05$.
*** $p < 0.001$.

(more than 10% for each category), polynomial regression and response surface analysis were conducted. The findings of the analysis indicated that there was a linear relationship between subjective family status and subjective school status. The findings revealed that as family and school statuses increased, anxiety levels decreased. As shown in Fig. 2, there was no significant difference in anxiety levels among individuals according to the congruency of family and school statuses.

## The mediating effect of anxiety and the moderating effect of psychological resilience

The results of analyses examining the mediating effect of anxiety and the moderating effect of psychological resilience are presented in Tables 3 and 4. Model 1 shows that sex, age,

**Table 5 Mediating effects of anxiety at different levels of mental resilience.**

| Dependent variable | Level | Mediation effect value | Bootstrap SE | 95% CI | Direct effect | Bootstrap SE | 95% CI | Proportion of mediating effect |
|---|---|---|---|---|---|---|---|---|
| | −1 level | 0.1 | 0.04 | [0.03–0.18] | 0.07 | 0.03 | [0.001–0.14] | 58.8% |
| Depression | Mean | 0.08 | 0.04 | [−0.01 to 0.16] | 0.07 | 0.03 | [0.02–0.13] | 53.3% |
| | 1 level | 0.12 | 0.03 | [0.06–0.18] | 0.07 | 0.03 | [0.03–0.12] | 63.15% |
| | −1 level | 0.16 | 0.06 | [−0.01 to 0.13] | 0.1 | 0.05 | [−0.01 to 0.20] | 43.8% |
| Suicidal attempts | Mean | 0.05 | 0.03 | [−0.004 to 0.11] | 0.1 | 0.04 | [0.02–0.17] | 33.3% |
| | 1 level | 0.07 | 0.03 | [0.03–0.12] | 0.09 | 0.03 | [0.03–0.15] | 43.75% |

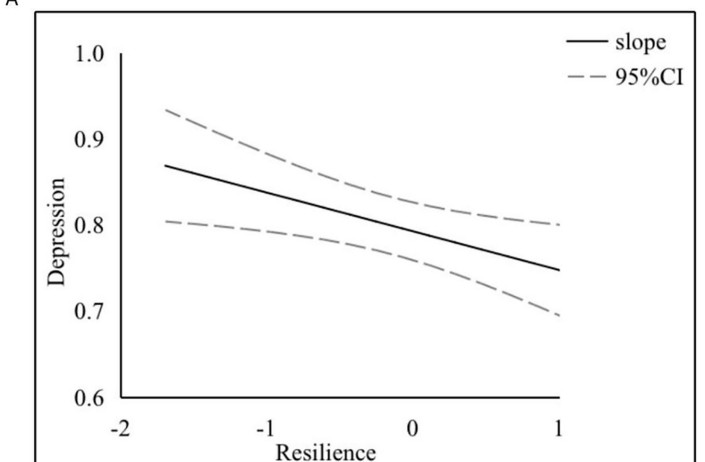 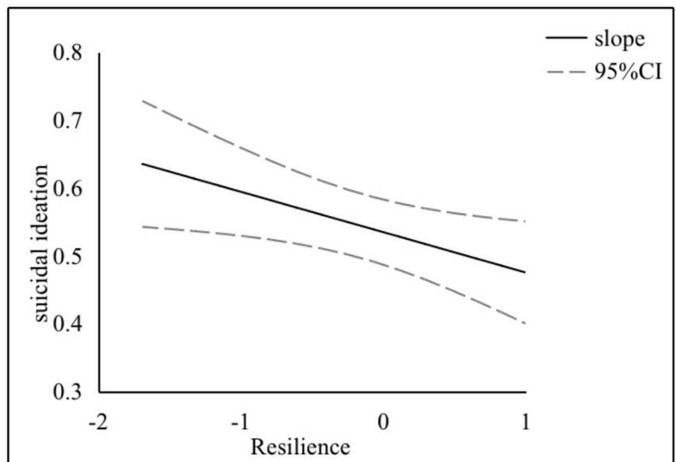

**Figure 3 The moderating role of resilience in the effects of anxiety on adolescent depression (top) and suicidal attempts (bottom) simple slope plot.** (A) The negative association between resilience and depression levels. (B) The relationship between resilience and suicidal ideation, with the solid line detailing the slope and the dashed lines indicating the 95% CI.

and the block variables significantly predicted adolescent depression and suicidal ideation ($\beta_{\text{depression}} = 0.11$, $p = 0.001$; $\beta_{\text{depression}} = 0.19$, $p < 0.001$; $\beta_{\text{depression}} = 0.21$, $p < 0.001$; $\beta_{\text{suicide attempts}} = 0.11$, $p < 0.001$; $\beta_{\text{suicide attempts}} = 0.17$, $p < 0.001$; $\beta_{\text{suicide attempts}} = 0.15$, $p < 0.001$). Model 2 shows that sex, age, and the block variables had significant effects on anxiety ($\beta = 0.08$, $p < 0.001$; $\beta = 0.17$, $p < 0.001$; $\beta_{\text{block variables}} = 0.15$, $p < 0.001$). Model 3 shows that sex, age, the block variables, and anxiety significantly predicted adolescent depression and suicide attempts ($\beta = 0.04$, $p < 0.001$; $\beta = 0.06$, $p < 0.001$; $\beta_{\text{block variables}} = 0.08$, $p < 0.001$; $\beta = 0.03$, p < 0.001; $\beta = 0.04$, $p < 0.001$; $\beta_{\text{block variables}} = 0.04$, $p < 0.001$; $\beta_{\text{anxiety}} = 0.20$, $p < 0.001$). Furthermore, the interaction terms of anxiety and resilience had significant effects on adolescent depression and suicide attempts ($\beta_{\text{depression}} = -0.07$, $p < 0.001$; $\beta_{\text{suicide attempts}} = -0.03$, $p < 0.05$). This suggests that the block variables (subjective family status and subjective school status), anxiety, adolescent depression, suicidal ideation, and psychological resilience constitute a moderated mediation model, with anxiety playing a partial mediating role and psychological resilience

moderating the second half of the mediating path, specifically the impact of anxiety on adolescent depression and suicidal ideation.

In addition, Table 5 lists the mediating effects of psychological resilience at the mean and ±1 standard deviation from the mean. In each regression model, with the increase in psychological resilience, the proportion of the mediating effect gradually decreased.

To clarify the moderating effect, a simple slope analysis was conducted to examine the role of psychological resilience in the impact of anxiety on adolescent depression and suicide attempts (see Fig. 3). The findings indicated that psychological resilience is a protective factor against adolescent depression and suicide attempts and can mitigate the impact of anxiety on these outcomes.

## DISCUSSION

This study use polynomial regression and response surface analysis based on the social causation hypothesis and diathesis-stress model to investigate the differential effects of subjective family status and subjective school status on adolescent depression and suicidal ideation. The results showed that as the economic status of the family and school increased, anxiety decreased. The differences in anxiety in individuals with high subjective family status and low subjective school status or low subjective family status and high subjective school status were not significant. This suggests that the effects of subjective family status and subjective school status on adolescent depression and suicidal ideation were not significantly different. In addition, this study identified the mediating and moderating effects of anxiety and psychological resilience, shedding light on the developmental mechanisms underlying the impact of subjective family and school circumstances on adolescent depression and suicidal ideation. These findings provide valuable insights regarding interventions targeting psychological crisis behaviors.

### Mediating effect of anxiety

This study used polynomial regression and response surface analysis to examine the joint impact of subjective family status and subjective school status on adolescent anxiety based on diathesis-stress theory. The results showed that both subjective family status and subjective school status had a negative effect on anxiety and could increase the risk of adolescent depression and suicidal ideation by increasing anxiety, indicating that anxiety played a mediating role. These findings are consistent with previous research showing that adolescents with high subjective family status and school status face less pressure and have lower risks of depression and suicidal ideation (*Dickerson, Milojevich & Quas, 2022*), while those with low subjective family status and school status are more likely to experience negative emotions, leading to depression, suicidal ideation, or more serious psychological disorders and diseases (*Ritterman, 2007*; *Zhou et al., 2021*). In addition, previous studies have mostly examined the effects of individual variables or risk factors in a single domain on adolescent mental health using simple total scores or weighting by a certain proportion, overlooking potential differences in the family and school environments. This study use polynomial regression and response surface analysis to explore the effects of congruency of subjective family status and subjective school status. We found that compared to

individuals with low family status and low school status, adolescents with high family status and high school status had lower levels of anxiety; incongruency of family status and school status (high/low or low/high) did not lead to significant differences in anxiety. Therefore, compared to solely focusing on family social status, researchers should pay attention to the joint influences of the school and family, as both are indispensable. Developing ecological theory proposes that children's development is influenced by multiple ecological subsystems, such as the family, school, and society, and the risks associated with different ecological subsystems interact with individual psychological factors, thereby affecting individual development (*Bronfenbrenner & Morris, 1998*). Children with high subjective family status may experience anxiety, depression, and even suicidal ideation if they do not perceive respect and care at school, and the same is true for adolescents with high subjective school status but low family status. Therefore, when investigating adolescent anxiety, researchers should not focus solely on the singular impact of either family social status or school status. Instead, they should consider the interaction between these two factors and their overall influence on individual mental health. This comprehensive approach allows for a more complete understanding of the complex mechanisms that affect adolescent mental health, thereby providing a more realistic and effective basis for developing intervention strategies. Such an integrative perspective is essential for understanding the multiple sources of adolescent anxiety and offers practical guidance for educators, parents, and policymakers, ultimately promoting the healthy development of adolescents on multiple levels. Additionally, the mediating effect of anxiety suggests that external pressures and stressful events may cause related psychological problems and disorders by inducing anxiety, which is consistent with previous research results (*Campisi et al., 2022*; *Capron et al., 2012*; *Zvolensky et al., 2018*).

### The moderating role of psychological resilience

The competence-stress model suggests that the development of psychological problems and disorders such as depression, anxiety, self-harm, and suicide are the result of the joint effects of external stressors and internal vulnerability factors. Psychological resilience, as a positive psychological resource that enables individuals to adapt to adversity and challenges, promotes individuals' mental and physical health and has an important impact on the development of adolescents (*Richardson, 2002*). This study confirmed that psychological resilience can buffer the negative effects of anxiety on depression and suicidal tendencies in adolescents. It had a moderating effect on the relationship of subjective family status and subjective school status with anxiety. The Johnson-Neyman plot shows that as psychological resilience increased, the impact of anxiety on adolescent depression and suicidal ideation gradually decreased. Additionally, the results in Table 5 show that with an increase in psychological resilience, the mediating effect of anxiety gradually weakened. All the above results indicate that psychological resilience can buffer the negative impact of anxiety. In a meta-analysis investigating the relationship between psychological resilience and various mental disorders, such as depression, emotional disorders, and bipolar disorder, *Imran et al. (2022)* reported that psychological resilience can alleviate emotional disorders in various populations, including pregnant women,

children, and adults, and have a positive effect on individuals' mental health. In studying psychological resilience, we found that it buffered impacts on depression and suicidal ideation; these findings may provide empirical guidance for interventions for depression and suicidal ideation among adolescents.

## Implications and limitations

Finally, it should be noted that this study used polynomial regression and response surface analysis to explore the effects and differences in subjective family status and subjective school status involvement experienced by adolescents. We also identified mediating and moderating effects of anxiety and psychological resilience, enriching understanding of the development of depression and suicidal ideation in adolescents and providing some guidance for future interventions. However, the cross-sectional design used in this study prevents us from determining causal relationships. Our project is still ongoing, and we will conduct longitudinal research on changes in adolescent mental health over multiple time points following this baseline survey. Additionally, one notable limitation of our study is the reliance on self-reported data collected through an online platform with parental supervision. Self-report measures are susceptible to biases such as social desirability bias and response distortion, where participants may alter their responses to align with perceived expectations or to avoid negative judgments. These factors could potentially affect the accuracy and reliability of the data. Future studies should consider incorporating additional data collection methods, such as interviews or objective measures, to complement self-reported data and mitigate these biases.

## CONCLUSIONS

Subjective family status and school status played important roles in depression and suicidal ideation in adolescents, and anxiety and psychological resilience played mediating and moderating roles. Specifically, the finding that interventions should not only address family and school environments but also focus on enhancing psychological resilience among adolescents. The theoretical significance of the identified mediating and moderating roles lies in expanding our understanding of how subjective perceptions of family and school environments interact with psychological factors to influence mental health outcomes. These findings provide a deeper understanding of the mechanisms underlying depression and suicidal ideation in adolescents.

## ACKNOWLEDGEMENTS

The authors are grateful to the participants for contributing to this research.

### Funding

This work was supported by the National Natural Science Foundation of China (82071543, 82171509), the Clinical Nursing Research Foundation of the Second Xiangya Hospital of Central South University (2021-HLKY-04), the Medical Research Foundation of Hunan

Medical Association (LYG2021096), the Key Guiding Project of Hunan Health Committee (202103091470), STI2030-Major Proiects-2021ZD0200700 and the Fundamental Research Funds for the Central Universities of Central South University (2022ZZTS0858). The funders had no role in study design, data collection and analysis, decision to publish, or preparation of the manuscript.

### Grant Disclosures

The following grant information was disclosed by the authors:
National Natural Science Foundation of China: 82071543, 82171509.
Clinical Nursing Research Foundation of the Second Xiangya Hospital of Central South University: 2021-HLKY-04.
Medical Research Foundation of Hunan Medical Association: LYG2021096.
Key Guiding Project of Hunan Health Committee: 202103091470.
STI2030-Major Proiects-2021ZD0200700 and the Fundamental Research Funds for the Central Universities of Central South University: 2022ZZTS0858.

### Competing Interests

The authors declare that they have no competing interests.

### Author Contributions

- Zhan Shu conceived and designed the experiments, performed the experiments, analyzed the data, prepared figures and/or tables, authored or reviewed drafts of the article, and approved the final draft.
- Shurui Chen conceived and designed the experiments, performed the experiments, analyzed the data, prepared figures and/or tables, authored or reviewed drafts of the article, and approved the final draft.
- Hui Chen performed the experiments, prepared figures and/or tables, and approved the final draft.
- Xianliang Chen analyzed the data, prepared figures and/or tables, and approved the final draft.
- Huajia Tang analyzed the data, prepared figures and/or tables, and approved the final draft.
- Jiawei Zhou analyzed the data, prepared figures and/or tables, and approved the final draft.
- Yusheng Tian analyzed the data, prepared figures and/or tables, and approved the final draft.
- Xiaoping Wang conceived and designed the experiments, authored or reviewed drafts of the article, and approved the final draft.
- Jiansong Zhou conceived and designed the experiments, authored or reviewed drafts of the article, and approved the final draft.

### Ethics

The following information was supplied relating to ethical approvals (*i.e.*, approving body and any reference numbers):

The ethics committee of the National Clinical Research Center in the second Xiangya Hospital approved the study (2018S007).

## Data Availability

The raw measurements are available in the Supplemental File.

## Supplemental Information

Supplemental information for this article can be found online at http://dx.doi.org/10.7717/peerj.18225#supplemental-information.

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
