# Peer review of "The effects of subjective family status and subjective school status on depression and suicidal ideation among adolescents: the role of anxiety and psychological resilience"

_PeerJ, doi:10.7717/peerj.18225_

## Round 0.1 · original submission · Minor Revisions

· Academic Editor

Minor Revisions

I've received two thoughtful reviews of the work, which are both quite positive but also raise some important issues that should be addressed prior to publication. These are mostly related to how the material is explained and interpreted, rather than requiring new analyses, so I'm issuing a decision of "Minor Revisions," but please consider these comments carefully as you prepare a revision. Depending on the responses, it may be necessary to send the manuscript out for re-review at that point.

Reviewer 1 ·

Basic reporting

No comment

Experimental design

No comment

Validity of the findings

No comment

Additional comments

The present study searches for the differential effects of subjective family status and subjective school status on adolescent depression and suicidal ideation, using polynomial regression and response surface analysis. Results indicate that higher subjective economic status reduces anxiety, which mediates the relationship between family/school status and mental health outcomes. Psychological resilience moderates these effects, buffering against anxiety-induced depression and suicidal ideation. The study is an original, innovative one with clearly written hypothesis and methodology. Article needs several revisions;
1. In the method, (lines 156-157) it has been indicated that participants were selected using a cluster sampling method. But there is not enough detail about how clusters were defined and selected. More information on the sampling process should be added.
2. More details are needed about inclusion criteria of the study. How mmany cases were approached , how many excluded why? As this study was a part of the China Depression Cohort Study (CDCS). Information about the parent study should be provided. What is the difference from original study? Only sample size, please elaborate on it.
3. Method section does not provide enough detailed information about the procedural steps taken during data collection. For instance, how were participants recruited? What specific measures were taken to ensure the reliability and validity of the self-reported data? What is the online platform (Wenjuanxing)? Information on how this platform ensures data integrity, security, and participant engagement is required.
4. More detail about development, validation, and reliability of measures used in the context of the current study is needed. More comprehensive descriptions of the measures, cut off points and their psychometric properties (original and adapted versions) should be added.
5. As stated, the study relies on self-reported data collected via an online platform with parental supervision. This clearly introduce biases. Authors should mention this issue as a limitation of the study.
6. The study should explain why polynomial regression and response surface analysis were chosen over others. Why other potential methods, such as structural equation modeling or hierarchical linear modeling, are not preferred? Please elaborate on it. Additionally, the steps taken to perform these analyses should be detailed. For example, what specific polynomial terms were included in the regression models? How were the response surfaces interpreted? More details are needed.
7. The statement indicating that this study is "the first to use polynomial regression and response surface analysis based on the social causation hypothesis and diathesis-stress model" may be overstated. This statement should be softened and authors should clarify how this study builds on or differs from previous research rather than implying complete novelty without sufficient justification.
8. There is a lack of critical analysis of the results in discussion. Discussion should address limitations or alternative explanations for the results. For example, the non-significant differences in anxiety between high and low subjective status groups are stated but not adequately explained or explored. This section should be revised from this perspective.

·

Basic reporting

The manuscript provides a clear and concise introduction and successfully contextualizes the research within the broader adolescent mental health literature. The focus on depression and suicidal tendencies in adolescents is extremely relevant given the increasing concern in this area. The introduction effectively illustrates the gap in understanding the effects of social status, which justifies the need for the present study. The background is well-referenced, with appropriate citations from contemporary and seminal literature supporting the study's rationale.
The literature review is thorough and relevant, drawing on various studies to support the hypotheses and research questions. The references are current and the authors have made a concerted effort to include both international and regional studies, adding depth to the discussion. The review is well structured and moves logically from general discussions of adolescent mental health to more specific topics such as social status, anxiety and psychological resilience.
The methods section is adequately detailed and contains clear descriptions of the instruments used (PHQ-8, GAD-7) and the sample population. The use of advanced statistical methods (Mplus, SPSS Process and RSA 3.0 plugin) is commendable and indicates a rigorous approach to data analysis. The choice of statistical tools is entirely justified as the studies focus on mediating and moderating effects. However, further details on the reasons for the selected age group and any possible biases in the sample selection process would increase the transparency of the study.
The results are presented and statistically reliable. The connections and mediating and moderating effects are appropriately analyzed and reported. The statistical significance of the results is clearly communicated and the results are presented in a manner that is accessible to readers with varying levels of statistical expertise. Using figures to illustrate key results is effective, although including a structural equation model diagram could further improve clarity.
The discussion section effectively interprets the results in the context of the existing literature and provides insightful reflections on the implications for adolescent mental health. The authors adequately discuss the limitations of their study, including possible biases and the generalizability of the results. The conclusions drawn are supported by the data and contribute significantly to the field, particularly to understanding the role of anxiety and psychological resilience.
The figures included in the manuscript are of high quality, well labelled and contribute to the understanding of the study results. The tables are clear and provide all the necessary information in a compact form.
The manuscript conforms to the structure expected in high-quality psychology journals such as PeerJ. The text is clear, professional and free of ambiguous expressions. The flow of the manuscript is logical, with each section building on the previous one. The study's objectives, methods, results, and conclusions are communicated effectively, making the manuscript accessible and informative to a broad readership.
Provide more detailed information about the reasons for the chosen age range and address possible sampling biases. Consider including a structural equation model diagram to visually represent the relationships between variables.
This manuscript represents a significant contribution to the literature on adolescent mental health, particularly in examining the role of social status, anxiety, and psychological resilience in depression and suicidal ideation. With minor revisions, it would be suitable for publication in PeerJ.

Experimental design

The study employs a robust design using widely accepted instruments (PHQ-8, GAD-7) and a significant sample size (N = 1,190) to assess depression, anxiety, and suicidal ideation in adolescents. Using Mplus, SPSS Process and RSA 3.0 for data analysis is appropriate. However, further clarity is needed on sampling methods and possible biases.

Validity of the findings

The results presented in the summary are supported by a large sample size and established measures such as the PHQ-8 and GAD-7. However, the report lacks details on the robustness of the statistical methods used and possible confounding factors. Further examination of these areas is required to confirm the validity of the conclusions

Additional comments

The study presents a well-structured examination of the relationship between social status, anxiety, psychological resilience, depression and suicidal ideation in adolescents. The study methodology, which includes large sample size and the use of established instruments such as the PHQ-8 and GAD-7, is robust. However, the summary could benefit from clarifying the practical implications of the results and further elaborating on the theoretical significance of the identified mediating and moderating roles. Overall, the study is relevant and timely.

---

## Round 0.2 · accepted · Accept

· Academic Editor

Accept

Both reviewers agree that the revised manuscript has adequately addressed all their comments and concerns, and that the article will make a nice contribution to this literature.

Reviewer 1 ·

Basic reporting

Authors responded all of my comments adequately

Experimental design

All done

Validity of the findings

No problem

·

Basic reporting

It is sufficient now after the revision.

Experimental design

It is sufficient now after the revision.

Validity of the findings

It is sufficient now after the revision.

Additional comments

It is sufficient now after the revision.